# High-Yield Preparation and Characterization of Feline Albumin with Antioxidant Properties and In Vivo Safety

**DOI:** 10.3390/ijms252313095

**Published:** 2024-12-05

**Authors:** Mengyin Deng, Quanlong Wu, Na Yang, Da Teng, Yexuan Wang, Ya Hao, Haiqiang Lu, Ruoyu Mao, Jianhua Wang

**Affiliations:** 1Gene Engineering Laboratory, Feed Research Institute, Chinese Academy of Agricultural Sciences, Beijing 100081, China; 2Innovative Team of Antimicrobial Peptides and Alternatives to Antibiotics, Feed Research Institute, Chinese Academy of Agricultural Sciences, Beijing 100081, China; 3Key Laboratory of Feed Biotechnology, Ministry of Agriculture and Rural Affairs, Beijing 100081, China; 4Enzyme Engineering Laboratory, College of Food Science and Technology, Hebei Agricultural University, Baoding 071001, China

**Keywords:** feline serum albumin, *Pichia pastoris*, high-level expression, antioxidant activity, in vivo safety

## Abstract

To address the limited supply of serum albumin (SA) and potential pathogen contamination, focus has been concentrated on the heterologous expression of human serum albumin (HSA), particularly in *Pichia pastoris*. However, there are rare studies on feline serum albumin (FSA), which requires a large amount in pet foods and clinical treatment. In this work, the codon-optimized recombinant feline serum albumin (rFSA) sequence was designed and transferred into *P. pastoris* GS115 for recombinant expression. The high-level expression strain was selected by a high concentration of G418, followed by plate and shaker screening. At the 5 L fermenter level, the total protein concentration reached 3.89 mg/mL after 113 h of induction. At lower concentrations (1–4 μM), rFSA exhibited a potent free radical scavenging capacity, reaching 99% and 60% for ABTS^+^• and •O^2−^, respectively, which surpassed that of natural plasma-derived FSA. The secondary structure and stability of rFSA were found to be consistent with those of FSA. Additionally, an in vivo safety assay in mice showed no significant difference between the rFSA group and the normal saline group in terms of body weight changes, complete blood count, serum biochemistry, inflammatory factors, and tissue sections. These results above indicate that *P. pastoris* is the optimal host for the high preparation of rFSA. Furthermore, rFSA has been demonstrated to be relatively safe, which paves the way for subsequent industrialized production and its application in pet foods and veterinary clinics.

## 1. Introduction

Serum albumin (SA) is the most abundant protein in vertebrate plasma, representing approximately 60% of the total plasma protein. The molecular mass of SA is approximately 66 kDa, and it consists of approximately 585 amino acids that have not undergone glycosylation. Additionally, SA comprises 17 conserved disulfide bridges and a free sulfhydryl group at the N-terminal 34th position. It performs many vital physiological functions, including maintaining the colloid osmotic pressure of plasma, facilitating the transportation of a range of substances, including hormones, amino acids, fatty acids, and divalent metal ions [1], and scavenging free radicals [2]. In human clinical settings, a high dose of SA is employed as a blood volume expander to augment blood volume levels in the aftermath of hemorrhagic shock, severe burns and traumas, and surgical procedures. Furthermore, it is utilized in the treatment of hypoalbuminemia resulting from ascites, liver cirrhosis, renal disease, hemorrhage, and other conditions. In the pharmaceutical industry, researchers fuse human serum albumin (HSA) with drugs to improve the blood half-life of therapeutic proteins [3], making it a nano-carrier for the targeted delivery of drugs [4,5] and an ingredient in vaccine formulations and cell culture media.

The number of pets has increased significantly from a relatively large base in recent years. Similarly, in China, the world’s second-largest pet market, there were over 121.55 million pet dogs and cats as of 2023 [6]. This has resulted in an increased demand for veterinary pharmaceuticals. In the absence of animal-specific serum albumin products in veterinary clinics, HSA is employed as a treatment for hypoalbuminemia in dogs and cats [7], albeit in small doses, which can precipitate allergic symptoms [8,9]. Furthermore, despite the sequence homology of feline serum albumin (FSA) and human serum albumin reaching 80%, there are still notable differences, such as a smaller Sudlow II site and a more flexible cysteine at position 34 in FSA, which result in FSA exhibiting distinct ligand-binding affinity [10].

Worldwide, the clinical therapeutic demand for HSA has reached hundreds of tons and continues to rise [11], posing challenges due to limited supply and the risk of pathogen infection associated with plasma-derived SA. To circumvent these difficulties, transgenic technology has been widely applied to express SA heterologously. Significant efforts have been made to enhance the expression of recombinant HSA (rHSA) in hotter and more mature conditions [12,13]. Nevertheless, there is a paucity of research concerning the large-scale and efficient production of recombinant FSA (rFSA).

The objective of this study is to investigate the feasibility of expressing rFSA in *P. pastoris*. The constructed rFSA plasmid vector was transferred into the host, and the highly resistant transformants were screened for high-density fermentation expression. The total protein concentration in the supernatant reached 3.89 g/L, which is a considerable level when compared to the expression of rFSA alone. The purified rFSA product was characterized in terms of its secondary structure, stability, in vitro antioxidant activity, and in vivo safety.

## 2. Results

### 2.1. Screening of Positive Transformants

To achieve the recombinant expression of FSA, the codon-optimized FSA gene sequence cleaved by the restriction enzymes *Eco*R I and Not I was ligated to the pPIC9K plasmid, resulting in the recombinant vector pPIC9K-rFSA (Figure 1A). The linearized pPIC9K-rFSA was transformed into *P. pastoris* GS115. Following two rounds of screening by an MD plate and antibiotic G418, fifty high-resistance positive transformants (2 mg/mL G418) were randomly selected for induction expression in a 48-well plate for 96 h. It was shown that the size of the protein band matched the expected theoretical value of 66 kDa (Figure 1C), and the concentrations of the target proteins were maintained at around 250 μg/mL in all cases.

### 2.2. Expression and Purification of rFSA

The highest-expressed transformant at the well plate level was subjected to 500 mL shake flasks and 5 L fermenter expression. As shown in Figure 1D, the total protein of the supernatant in the 5 L fermenter exhibited a gradual increase, reaching 3.89 g/L at the end of fermentation (113 h of methanol induction). Concurrently, the total biomass of GS115-pPIC9K-rFSA increased to 422.93 g/L. The sodium dodecyl sulfate polyacrylamide gel electrophoresis (SDS-PAGE) analysis (Figure 1E) showed that rFSA (about 66 kDa) became more abundant with the prolongation of the induction period. The rFSA was present in a higher proportion relative to the degradation products (about 45 kDa) in the supernatant. Subsequently, the fermentation supernatant was purified twice by a Blue chromatography column, and the final purified product was observed as a single band by SDS-PAGE, with 96% purity as determined by reversed-phase high-performance liquid chromatography (RP-HPLC) (Figure 1F).

### 2.3. Free Sulfhydryl Content

The quantification of free sulfhydryl groups was necessary to determine whether the disulfide bonds in rFSA would be mispaired during the secretory expression in *P. pastoris*. The results demonstrated that rFSA and natural plasma-derived FSA contained comparable total free sulfhydryl groups per mole, with values of 1.01 mol and 1.16 mol, respectively, aligned with the theoretical values.

### 2.4. Secondary Structure 

The secondary structure of rFSA was analyzed via circular dichroism (CD). As shown in Figure 2, the CD spectra of rFSA and FSA exhibited high overlap, with the two characteristic negative peaks of α-helix appearing near 208 nm and 222 nm. Specifically, the CDNN software version 2.1 calculated consistent results regarding the percentage of secondary structure elements, with the helical structure accounting for the majority (Table 1). These indicated that the expressed rFSA was correctly folded.

### 2.5. Stability

To evaluate the stability of rFSA, the rFSA was treated under a range of conditions and a grayscale analysis was conducted to ascertain its retention, taking the retention of untreated samples as 100%. On the whole, compared with FSA, rFSA showed comparable stability at various temperatures (Figure 3A) and pH (Figure 3B), with no significant differences observed. Specifically, the proteins were stable in the range of 25 °C to 70 °C, with retention rates exceeding 80%. When the temperature reached 85 °C, the proteins exhibited a notable degradation, and even at 100 °C, only about 10% were retained. The proteins demonstrated high retention within the pH range of 1 to 11, with optimal stability observed at pH 7. However, in an environment with a highly alkaline pH of 13, there was only 50% remaining. For plasma stability, rFSA also had a similar retention profile to FSA, demonstrating minimal degradation even after 480 min of prolonged incubation (Figure 3C).

### 2.6. In Vitro Antioxidant Activity

To determine whether there is a difference in the antioxidant capacity of rFSA and FSA, their relative total antioxidant capacity was measured by a colorimetric assay in addition to their capacity to scavenge two synthetic free radicals (ABTS^+^• and DPPH•) and two naturally occurring free radicals in living organisms (OH• and •O^2−^). As shown in Figure 4A, the total antioxidant capacity per mole of FSA and rFSA was equivalent to 0.39 and 0.43 mole of Trolox, respectively. The overall trend in the scavenging effects of rFSA and FSA on the four free radicals exhibited a consistent pattern with a dose-effect relationship, although some differences were observed. At lower concentrations (1–4 μM), rFSA demonstrated a greater inhibitory effect on ABTS^+^• and •O^2−^ than FSA (Figure 4B,E). Similarly, at a higher concentration of 16 μM, rFSA inhibited DPPH• and OH• at a higher rate than FSA (Figure 4C,D).

### 2.7. Hemolysis

FSA is mainly administered via intravenous injection, so a hemolysis test is necessary to ascertain whether the rFSA is toxic to red blood cells. It is indicated that the hemolysis rate of rFSA was nearly 1% within the concentration range of 5–160 μM (Figure 5A), thereby confirming that the rFSA possessed good hemocompatibility. The safety of rFSA in mice could be further investigated by intravenous injection.

### 2.8. In Vivo Safety Assay

From the initial day of tail vein injection to the seventh day, it was observed that the weight of the mice had been increasing steadily (Figure 5B) and no abnormal reaction was observed. After one week of continuous injection, there were no statistically significant differences in whole blood test indices, serum biochemical indices, antioxidant indices, and immune factors in the rFSA group compared with the control group (Table 2 and Table 3). Tissue sections revealed no evidence of abnormalities or lesions in the hearts, livers, spleens, and kidneys of the mice, and there were no histomorphological differences compared with the control group (Figure 5C). In conclusion, the administration of rFSA via the intravenous route is associated with a favorable safety profile.

## 3. Discussion

There is a substantial body of literature that documents the expression of SA in microbial systems. Reininger et al. successfully obtained intact FSA cDNA from cat hepatocyte RNA by RT-PCR and transfected it into *E. coli* for expression with yields of 2–10 mg/L [14]. However, recombinant proteins expressed at high levels in *E. coli* tend to aggregate to form insoluble inclusion body particles [15]. These particles must be isolated from the cells, solubilized, refolded, and undergo other treatments to produce active recombinant proteins [16], which undoubtedly increases the purification cost. Consequently, researchers have endeavored to employ fusion expression techniques and genetically modified strains to enhance protein solubility and facilitate the formation of disulfide bonds in the cytoplasm. Nguyen et al. expressed human protein disulfide isomerase and maltose-binding protein (MBP) fused with rHSA in an engineered strain, Origami 2, to make it soluble, resulting in a significant increase in solubility of more than 90%, with a final yield of 18.92 mg/L culture [17]. This indicates that the *E. coli* system still suffers from low yield. Conversely, the *P. pastoris* expression system offers the same benefits as other yeast expression systems, including rapid growth and the capacity to perform post-translational modifications (glycosylation and disulfide bond formation). Additionally, it has the advantages of low endogenous protein levels, high secretory expressivity, and a lower degree of high mannosylation [18]. Mallem et al. selected *P. pastoris* mut^s^ as the expression strain and optimized the production processes such as cell density, growth rate, and temperature to increase the expression of rHSA to 10 g/L (about 400 h induction) [19]. Zhu et al. performed rHSA expression by highly resistant transformants at levels of up to 8.86 g/L (96 h induction) [20]. Increasing the number of cognate genes, i.e., gene copies, typically results in an elevated expression of the target gene [21]. In this work, highly resistant transformants (typically exhibiting high copy numbers) were screened for high-density expression using the antibiotic G418, resulting in an rFSA yield of 3.89 g/L (113 h induction), which was 33.28-fold higher than that observed in the previous study [22]. Meanwhile, the exact copy number requires further determination.

In general, recombinant serum albumin undergoes degradation during the latter stages of high-density fermentation in *P. pastoris*, producing a degradation fragment of about 45 kDa, which is also present in yeast cell lysate [23]. This phenomenon is mainly caused by the action of endogenous proteases A and B, as well as carboxypeptidase Y, which are released extracellularly due to massive cellular rupture during the latter stages of induction. The primary strategies for preventing the degradation of recombinant serum proteins are the optimization of the fermentation environment and process parameters and the utilization of protease-deficient strains. In general, lower temperatures [24] and pH levels [25,26] are conducive to protein folding and reduced protease activity. Additionally, various supplements, including amino acids [23], peptides (such as casein hydrolysate, peptone, and yeast extracts) [27], and other substances, can be incorporated to enhance the process. Strains lacking proteases (SMD1168, SMD1163, etc.) exhibit reduced growth rates and lower recombinant protein expression due to the absence of specific proteases. Accordingly, in this study, the fermenter was supplemented with peptone, yeast extract, and histidine in a staged manner to enhance yeast biomass and optimize protein expression. Peptides can serve as both a substrate for protease and a supplementary nitrogen source, while histidine can mitigate the burden of the His-deficient strain GS115 to synthesize His independently. Accordingly, the proportion of the target protein exceeded 75% (Figure 1E), and only a negligible quantity of degraded protein was detected.

The purification methods for recombinant serum albumin involve a combination of pretreatment involving heat, ammonium sulfate precipitation [28] or ultrafiltration [29], and subsequent fine separation through multistep chromatography. However, these methods are time-consuming or result in a significant loss of the target protein. The rHSA was adsorbed onto a NiNTA column, followed by the addition of urea to denature the protein and release internal impurities, ultimately leading to the crystallization of rHSA. Although the resulting protein crystals exhibited a high purity of over 98%, the recovery efficiency was only approximately 23% [30]. In this work, the supernatant was directly passed through Blue column chromatography. This eliminated some preliminary steps that were cumbersome and reduced the loss of protein [19,20,31]. In comparison with the target serum albumin, the medium of the Blue column demonstrated a more pronounced capacity for the adsorption of 45 kDa degradation fragments, thereby facilitating the separation of these fragments from the target proteins. Moreover, some studies have employed a combination of multiple chromatographic methods, including the Blue column in conjunction with a hydrophobic column [27] and anionic column [32], with the objective of enhancing the purity of the resulting product, and the protein recovery rate was consistently maintained at around 60%.

SA is the major endogenous substance exerting antioxidant effects in plasma [33], and its free thiol group at the N-terminal Cys-34 is the predominant source of plasma thiols [2]. In addition to the commonly recognized free cysteine-34, it has been demonstrated that methionine in SA is also involved in antioxidant activity [34,35,36]. Some abnormal chemical modifications, such as non-enzymatic glycosylation [37,38,39] and disulfide bond formation (-SH depletion of Cys-34) [40], can impair the antioxidant activity of albumin. In this study, the in vitro antioxidant activities of rFSA and FSA were evaluated. Given that the -SH contents of both were essentially identical, we postulated that the residual impurities in the purified products and certain modifications in rFSA were responsible for the relatively minor discrepancies in their free radical scavenging capabilities. Accordingly, further testing is required to ascertain whether rFSA is identical to FSA in structure and physiological function.

The clinical dose of SA is typically in the tens of grams, which necessitates the removal of immunoreactive impurities to ensure the highest possible degree of purity. In the absence of such purification, trace quantities of impurities may gain access to the patient’s body, potentially eliciting an adverse reaction [41,42]. Despite the 99.9% purity of rHSA extracted from rice endosperm, it remains hypoallergenic in mice [43]. Conversely, some studies have demonstrated that high-purity rHSA (yeast-expressed) is safe, with similar tolerability, immunogenicity, and PK/PD profiles as plasm-derived HSA [44,45]. The in vivo safety assay in mice revealed no significant alterations in hematological parameters, the absence of increased metabolic burden on the liver and kidneys, and the absence of systemic inflammation (Figure 5). However, despite the high degree of sequence homology observed among mammalian serum albumin, there are specific non-conserved regions that persist, potentially leading to divergent responses upon injection into different animal species. Therefore, a further safety evaluation in target animal cats will be conducted comprehensively.

## 4. Materials and Methods

### 4.1. Expression and Purification of rFSA

The sequence of FSA (GenBank: CAD32275.1) was optimized in accordance with the codon preference of *P. pastoris* using the GenSmart™ codon optimization tool. It was then inserted into the pPIC9K plasmid, thereby constructing the recombinant expression vector pPIC9K-rFSA, which was amplified in *E. coli* TOP10. The recombinant plasmid linearized by *Pme* I was transformed into the *P. pastoris* GS115 strain by electroporation and then coated on MD plates. Following 3–5 days, the transformants were picked and inoculated onto YPD plates containing different concentrations of G418 (0.5, 1, and 2 mg/mL) to facilitate the further screening of highly resistant transformants. The highly resistant transformant with the highest expression of the target proteins in 48-microtiter plate fermentation was selected for high-density fermentation in 5 L fermenters, as in previous method [20,27], with some modifications as follows: 20 g peptone, 10 g yeast extract, and 0.3 g histidine were added at 24 h intervals for a total of four times during the induction period.

The fermentation broth’s supernatant collected by centrifugation (4000 rpm, 30 min, 4 °C) was filtered through a 0.45 μM microfiltration membrane. Subsequently, the supernatant was diluted tenfold in buffer A (20 mM sodium phosphate buffer, pH 5.0) and purified using HiTrap Blue HP (Cytiva, Uppsala, Sweden). In detail, buffer A was used to pre-equilibrate the affinity chromatography column for up-sampling, buffer B (20 mM sodium phosphate buffer, 2 M NaCl, pH 6.8) was employed to elute the rFSA bound to the column, and the column was finally washed with 0.1 M NaOH. The collected eluate was subjected to dialysis in buffer A to desalt it and adjust the pH to 5.0, thus preparing it for a second purification step, as described above. The purified products were subjected to analysis by 12% SDS-PAGE and RP-HPLC (Agilent Technologies, Santa Clara, CA, USA).

### 4.2. Determination of Free Sulfhydryl Content

The reaction of free sulfhydryl groups with 5,5′-dithiobis (2-nitrobenzoic acid, DTNB) produces the yellow 2-nitro-5-thiobenzoate anion (TNB), which exhibits a pronounced absorbance peak at 412 nm. Briefly, different concentrations of L-Cys solution (5–160 μM, 125 μL) and rFSA solution (40 μM, 125 μL) were each mixed with a DTNB solution (1 mg/mL, 25 μL), and the absorbance at 412 nm was measured after 20 min of reaction at room temperature. A standard curve was established by the absorbance values of L-Cys, and the results obtained after substituting the absorbance values of rFSA into the curve were expressed as the number of free cysteine residues per mole of rFSA.

### 4.3. Secondary Structure Analysis

The secondary structures of rFSA and FSA (purified from feline serum, purity ≥ 97%) were analyzed via a Pistar π-180 circular dichroism (Applied Photophysics, Surrey, UK). Protein samples were solubilized by PBS to a concentration of 400 μg/mL. The solution was introduced into a 1 mm quartz cuvette for circular dichroism scanning within the range of 190–260 nm, with PBS employed as the blank control. The data were subsequently processed using Pro-Data Viewer and CDNN.

### 4.4. Temperature, pH, and Plasma Stability

The relative quantification of rFSA retained after treatment with different conditions was performed by SDS-PAGE and ImageJ.JS to evaluate the stability of rFSA. In the temperature stability test, the rFSA solution (4 μM in PBS) was incubated at 25 °C, 40 °C, 55 °C, 70 °C, 85 °C, and 100 °C for 1 h. In the pH stability test, the rFSA solution (40 μM in PBS) was mixed with phosphate buffer of pH 1, 3, 5, 7, 9, 11, and 13 at a ratio of 1:9 and incubated at 25 °C for 3 h. For the plasma stability test, the rFSA solution (40 μM in PBS) was mixed with mouse plasma at 3:1 and incubated for 15, 30, 60, 120, 240, and 480 min.

### 4.5. Antioxidant Activity In Vitro

#### 4.5.1. Total Antioxidant Capacity (T-AOC)

The T-AOC test was conducted according to the instructions provided in the kit with the FRAP method (Solarbio, Beijing, China). A standard curve was constructed by the absorbance values of the antioxidant Trolox, and the total antioxidant capacity of rFSA and FSA was expressed as the equivalent Trolox.

#### 4.5.2. DPPH• Scavenging Ability

The hydrogen atoms of antioxidants were paired with single electrons of DPPH• to reduce it to DPPH-H, with a decrease in the absorbance value at 517 nm. DPPH (0.2 mM, 125 μL) and the rFSA solution (1–32 μM, 125 μL) with different concentration gradients were added to a 96-well plate, and the absorbance was measured after incubation for 30 min at room temperature.

#### 4.5.3. ABTS^+^• Scavenging Ability

ABTS is oxidized to form the stable blue-green cation radical ABTS^+^•. The antioxidants reduce ABTS^+^• to colorless ABTS, causing a decrease in absorbance at 405 nm. ABTS (7.4 mM) was mixed 1:1 with K_2_S_2_O_8_ (2.6 mM), and the reaction was shielded from light for 12 h. The ABTS working solution (OD_405 nm_ ≈ 0.70 ± 0.02) was diluted approximately 50-fold with PBS. A mixture of the working solution (160 μL) and the rFSA solution (2.5–80 μM, 40 μL) stood for 5 min in the absence of light. The absorbance was read at 405 nm.

#### 4.5.4. OH• Scavenging Ability

The OH• generated by the reaction between H_2_O_2_ and Fe^2+^ oxidizes the color developer TMB to blue TMB^+^, which will be further oxidized to yellow TMB^2+^ under strong acidic conditions. The scavenging of OH• by antioxidants reduces the final yellow TMB^2+^ and the absorbance at 450 nm. FeSO_4_ (3 mM, 35 μL), H_2_O_2_ (3 mM, 35 μL), TMB (6 mg/mL, 10 μL), and the rFSA solution (2.5–80 μM, 20 μL) were added sequentially to a 96-well plate and reacted for 20 min. The absorbance was measured immediately after stopping the reaction with a 10% acetic acid solution (200 μL).

#### 4.5.5. •O^2−^ Scavenging Capacity

Xanthine oxidase catalyzes the formation of uric acid from xanthine in a reaction that produces •O^2−^, which reduces NBT to blue formazan absorbed at 560 nm. Antioxidants scavenge •O^2−^ and thus inhibit the formation of NBT formazan. The mixed solution of rFSA (60 μL), NBT (0.8 mM, 40 μL), xanthine (0.8 mM, 40 μL), and xanthine oxidase (10 U/mL, 10 μL) was incubated at 37 °C for 30 min, and then the absorbance was measured.

The scavenging rate of the four free radicals was calculated as follows: scavenge (%) = (1 − (A_sample_ − A_sample control_)/A_control_) × 100. A_sample_ is the absorbance of the reaction between the sample solution and each reagent solution, A_sample control_ is the absorbance of the sample solution itself, and A_control_ is the absorbance of the sample solvent and each reagent solution.

### 4.6. Hemolysis Test

Blood was collected from mice, subjected to centrifugation to remove the plasma, and washed with PBS until the supernatant was clear. An 8% erythrocyte solution (diluted in PBS) was mixed with an equal volume of the rFSA solution at a final concentration of 5–160 μM, and the mixture was incubated at 37 °C for 1 h. The supernatant was then centrifuged to determine the absorbance value at 540 nm (Ai). PBS and 0.1% Triton X-100 were used as the blank control (A0) and positive control (A1), respectively.
Hemolysis rate (%) = [(Ai − A0)/(A1 − A0)] × 100.

### 4.7. Acute Toxicity Assessment

Eighteen 6-week-old ICR female mice were randomly assigned to two groups and administered 200 μL of rFSA (150 mg/kg body weight) and normal saline via the tail vein for seven consecutive days. The weight of each mouse was recorded daily. On day 8, blood samples were collected for the purpose of conducting complete blood counts and serum biochemistry tests. The antioxidant indices of each serum sample were determined using the corresponding assay kit (Beyotime, Shanghai, China). The levels of inflammatory factors in serum were quantified using enzyme-linked immunosorbent assay kits (MEIMIAN, Yancheng, China). Furthermore, the heart, liver, spleen, and kidneys were subjected to histopathological examination to observe pathological alterations.

### 4.8. Statistical Analysis

All experimental data were analyzed using GraphPad Prism 8.0, and the *t*-test (α = 0.05) was performed to compare between groups. The results were shown as mean ± standard deviation (SD). A *p*-value of less than 0.05 was the criterion for assessing significant differences.

## 5. Conclusions

In this study, we constructed and screened the *P. pastoris* GS115 strain with a high expression of rFSA. A total supernatant protein concentration of 3.89 g/L was achieved at the 5 L fermenter level, with purification by simple two-time Blue column chromatography yielding high rFSA protein purity and recovery. The rFSA fermentation and purification processes in this study demonstrated potential for industrialization. rFSA products exhibited a similar secondary structure, stability, and in vitro antioxidant activity to plasma-derived FSA. The acute toxicity test in mice showed that rFSA has an acceptable safety profile, which makes it promising for applications in clinical studies in cats.

## Figures and Tables

**Figure 1 ijms-25-13095-f001:**
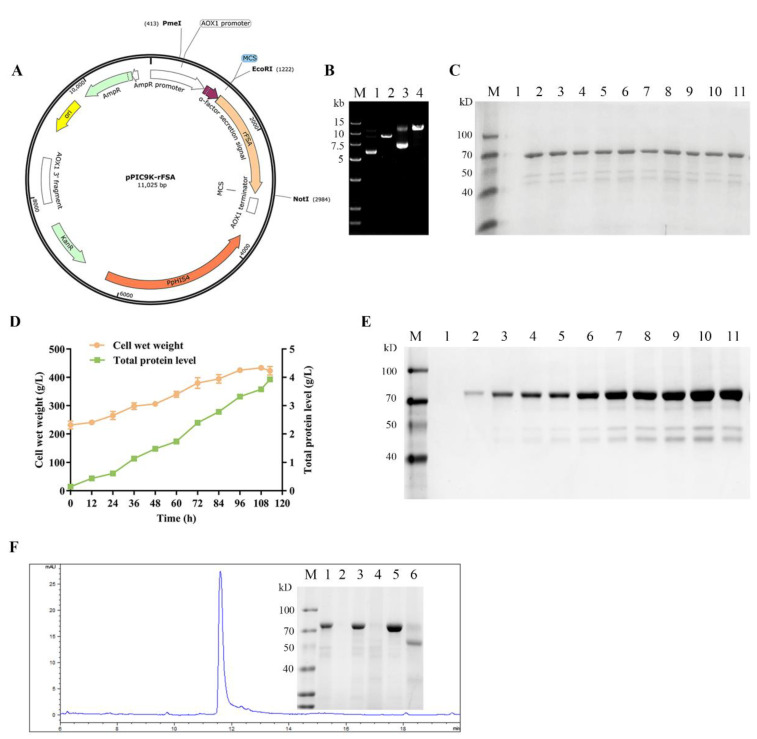
The screening of pPIC9K-rFSA transformants and the expression of rFSA in a 5 L fermenter. (**A**) A schematic map of the recombinant plasmid pPIC9K-rFSA. (**B**) Gel electrophoresis analysis of the circular and linearized pPIC9K-rFSA. Lane 1–2: circular and linearized pPIC9K; lane 3–4: circular and linearized pPIC9K-rFSA. (**C**) SDS-PAGE analysis of rFSA expression of high-resistance transformants. Lane 1: fermentation supernatant of transformant with pPIC9K; lanes 2–11: fermentation supernatant of transformants with pPIC9K-rFSA. The fifth transformant from left to right exhibited the highest protein concentration of 276 μgg/mL. (**D**) Curves of total supernatant protein and cell wet weight over time during the 5 L fermentation process. (**E**) SDS-PAGE analysis of rFSA expression at the 5 L fermenter level. Lanes 1–11: tenfold diluted fermentation supernatants at 0, 12, 24, 36, 48, 60, 72, 84, 96, 108, and 113 h induction. (**F**) RP-HPLC analysis of purified rFSA and SDS-PAGE analysis of rFSA purification. Lane 1: the final fermentation supernatant; lane 2: the penetrating fraction of the supernatant; lanes 3–4: the elute with 100% buffer B and 0.1 M NaOH in the 1st round; lanes 5–6: the elute with 100% buffer B and 0.1 M NaOH in the 2nd round.

**Figure 2 ijms-25-13095-f002:**
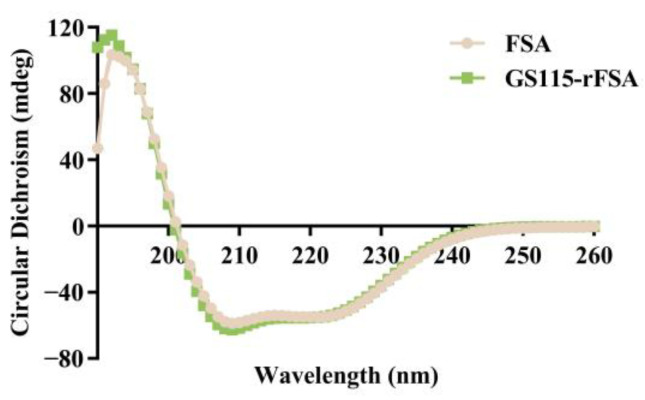
CD spectra of rFSA and FSA in PBS.

**Figure 3 ijms-25-13095-f003:**
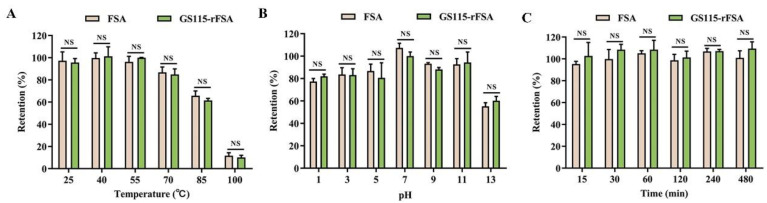
The stability of rFSA and FSA at different temperatures (**A**), pH (**B**), and incubation times with plasma (**C**). Data from three biological replicates were shown as the means ± SD. NS (*p* > 0.05) indicates no significant difference between rFSA and FSA.

**Figure 4 ijms-25-13095-f004:**
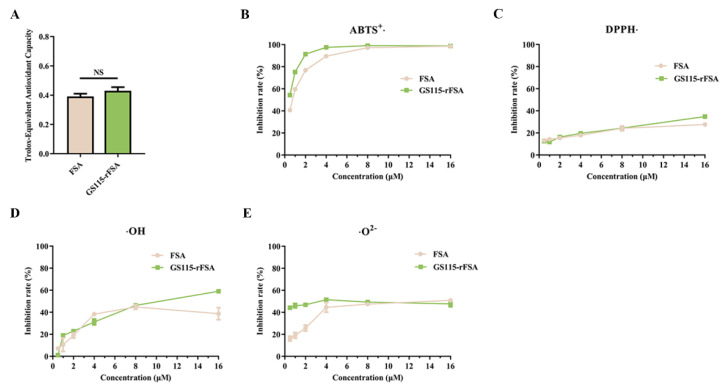
An antioxidant effect analysis of rFSA and FSA. The Trolox-equivalent antioxidant capacity (**A**) and inhibition rate of ABTS^+^• (**B**), DPPH• (**C**), OH• (**D**), and •O^2−^ (**E**) were obtained from the spectrophotometric data of three replicate experiments. NS (*p* > 0.05) indicates no significant difference between rFSA and FSA.

**Figure 5 ijms-25-13095-f005:**
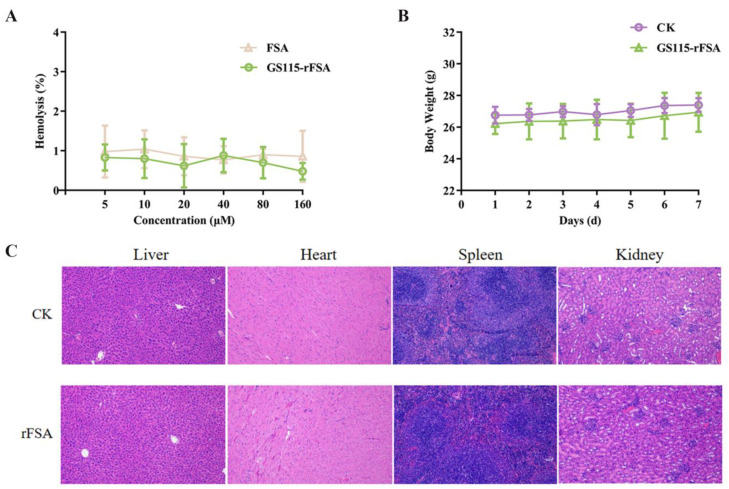
The safety evaluation of rFSA. (**A**) The hemolytic activity of rFSA and FSA (5–160 μM) in a 4% mouse erythrocyte solution. Body weight changes (**B**) and tissue sections (100×) (**C**) of mice after administering rFSA intravenously for 7 d. The normal saline group was used as the blank control.

**Table 1 ijms-25-13095-t001:** The percentages of the secondary structure elements of rFSA and FSA.

Second Structure	Wavelength 190–260 nm (%)	
	FSA	GS115-rFSA
Helix	51	52.5
Antiparallel	4.8	4.6
Parallel	5.6	5.3
Beta-Turn	14.1	13.8
Rndm Coil	23.2	22.4

**Table 2 ijms-25-13095-t002:** The hematological parameters of mice after the administration of rFSA intravenously for 7 d.

Item	CK	GS115-rFSA	*p*-Value
WBC (10^9^/L)	6.04 ± 4.07	5.07 ± 0.62	>0.05
LYM (10^9^/L)	1.09 ± 0.78	1.25 ± 0.36	
MID (10^9^/L)	0.48 ± 0.54	0.59 ± 0.18	
GRA (10^9^/L)	3.35 ± 2.23	2.85 ± 0.85	
PLT (10^9^/L)	657.33 ± 390.88	638.67 ± 208.40	
RBC (10^12^/L)	6.14 ± 0.32	5.20 ± 0.68	
HGB (g/L)	126.50 ± 6.36	114.50 ± 13.44	
HCT (L/L)	0.23 ± 0.01	0.21 ± 0.04	
MCV (fL)	37.80 ± 0.70	37.97 ± 1.80	
MCH (pg)	21.73 ± 2.47	20.93 ± 0.78	
MCHC (g/L)	547.00 ± 72.33	525.00 ± 42.79	

**Table 3 ijms-25-13095-t003:** The serum biochemical indices of mice including liver and kidney function, antioxidant capacity, and inflammatory factors after the administration of rFSA intravenously for 7 d.

Item	CK	GS115-rFSA	*p*-Value
ALT (U/L)	38.8 ± 4.3	34 ± 6.7	>0.05
AST (U/L)	180.3 ± 15.3	168.5 ± 20.6	
UREA (mmol/L)	7.6 ± 0.2	6.5 ± 1.1	
TP (g/L)	53.3 ± 3.9	50.8 ± 2.4	
ALB (g/L)	35.2 ± 1.5	32.2 ± 2.5	
T-AOC (mmol/L)	1.60 ± 0.28	1.77 ± 0.26	
CAT (U/mL)	107.42 ± 34.54	101.58 ± 36.46	
GSH-Px (U/mL)	424.09 ± 8.32	413.53 ± 11.14	
SOD (U/mL)	27.93 ± 5.22	27.74 ± 5.87	
MDA (nmol/L)	13.91 ± 0.68	13.37 ± 0.51	
IL-1β (pg/mL)	151.69 ± 15.30	156.20 ± 20.28	
IL-6 (pg/mL)	177.68 ± 13.79	185.87 ± 18.40	
IL-10 (ng/mL)	1.57 ± 0.20	1.58 ± 0.20	
TNF-α (ng/mL)	1.54 ± 0.10	1.57 ± 0.10	

## Data Availability

The original contributions presented in the study are included in the article; further inquiries can be directed to the corresponding author(s).

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
