# Peer review of "High-Yield Preparation and Characterization of Feline Albumin with Antioxidant Properties and In Vivo Safety"

_ijms, 2024, doi:10.3390/ijms252313095_

Round 1

Reviewer 1 Report

Comments and Suggestions for Authors

M. Deng et al., reported that codon-optimized recombinant feline serum albumin (rFSA) protein was successfully expressed in Pichia pastoris, and showing high yield, antioxidant capacity, structural stability, and safety for industrial applications.  Nonetheless, the paper is well written, and the experimental part is well-done, and the work should be published following the following 'major' considerations:

Major comment-1:

The authors have used the artificial radical molecules DPPH and ABTS to evaluate the antioxidant activity of the rFSA protein, but there are no positive control experimental data using well-known antioxidants (e.g., glutathione, ascorbic acid, Trolox). To evaluate the relative antioxidant activity of the authors' rFSA protein, it is necessary to develop a standard curve of reference substances and evaluate the antioxidant capacity.

Major comment-2:

The authors used TMB2+, which is produced by the reaction with hydroxyl radicals, to evaluate the hydroxyl radical scavenging capacity by measuring the absorbance at 450 nm, but in this experiment, it is impossible to deny the possibility that the fFSA protein is directly reacting with H2O2. The authors need to evaluate whether or not the rFSA protein reacts directly with H2O2, and then consider the effects of this. In addition, in this experiment, it is also necessary to use standard substances known to have anti-fenton reaction effects, such as ascorbic acid, to evaluate the antioxidant activity in a relative manner.

Major comment-3:

The authors have evaluated the superoxide radical scavenging capacity using the xanthine oxidase-mediated superoxide radical production system and NBT, but it has been reported that NBT interacts with xanthine oxidase. Therefore, the authors need to confirm that the rFSA protein does not directly interact with NBT, and also need to evaluate the superoxide radical scavenging capacity using other indicators such as XTT. In addition, in this experiment, it is also necessary to evaluate the antioxidant activity relative to standard substances known to have anti-fenton reaction effects, such as ascorbic acid, using antioxidants such as Trolox and SOD.

Minor comment: There are many typos and some abbreviations, the full name of which is not mentioned in the manuscript at the first appearance.  The authors should carefully check the typos and the grammar thoroughly.

M. Deng et al., reported that codon-optimized recombinant feline serum albumin (rFSA) protein was successfully expressed in Pichia pastoris, showing high yield, antioxidant capacity, structural stability, and safety for industrial applications.  The paper is well-written, and the experimental work is robust. The manuscript should be published after addressing the following major considerations:

Major Comment-1:

The authors evaluated the antioxidant capacity of the rFSA proteins using artificial radicals DPPH and ABTS.  However, no positive control experimental data using well-known antioxidants (e.g., glutathione, ascorbic acid, or Trolox) were provided.  To accurately assess the antioxidant capacity of rFSA, it is necessary to prepare a standard curve using reference antioxidant substances and evaluate its antioxidant capacity accordingly.

Major Comment-2:

The authors assessed the hydroxyl radical scavenging capacity using TMB2+, generated through a reaction with hydroxyl radicals produced by Fenton reaction, by measuring absorbance at 450 nm.  However, this experiment does not rule out the possibility that rFSA directly reacts with H2O2.  The authors should confirm whether rFSA reacts directly with H2O2, and if so, consider its implications. Additionally, this experiment should include standard antioxidant substances such as ascorbic acid or Trolox to evaluate antioxidant activity in a relative manner.

Major Comment-3:

The authors evaluated the superoxide radical scavenging capacity using the xanthine oxidase-mediated superoxide production system and NBT.  However, it has been reported that NBT directly interacts with xanthine oxidase.  The authors need to confirm that rFSA does not directly interact with NBT.  Furthermore, the superoxide radical scavenging capacity should also be evaluated using alternative indicators such as XTT.  In addition, the experiment should include relative evaluations using standard antioxidants such as Trolox and SOD.

Minor Comment:

There are numerous typos and instances where abbreviations are used without providing their full form upon first mention.  The authors should carefully proofread the manuscript for typos, grammar issues, and incomplete definitions of abbreviations.

Author Response

Comments 1: The authors have used the artificial radical molecules DPPH and ABTS to evaluate the antioxidant activity of the rFSA protein, but there are no positive control experimental data using well-known antioxidants (e.g., glutathione, ascorbic acid, Trolox). To evaluate the relative antioxidant activity of the authors' rFSA protein, it is necessary to develop a standard curve of reference substances and evaluate the antioxidant capacity.

Answer 1: Thank you for your comments. The objective of the antioxidant experiments was to ascertain whether there was a difference in the antioxidant capacity of rFSA and FSA. As you have suggested, it is imperative to assess the relative antioxidant capacity of rFSA in comparison with well-established antioxidants. Consequently, the Trolox was selected for supplementation. The total antioxidant capacity per mole of FSA and rFSA was equivalent to 0.39 and 0.43 mole of Trolox, respectively.

Comments 2: The authors used TMB2+, which is produced by the reaction with hydroxyl radicals, to evaluate the hydroxyl radical scavenging capacity by measuring the absorbance at 450 nm, but in this experiment, it is impossible to deny the possibility that the rFSA protein is directly reacting with H2O2. The authors need to evaluate whether or not the rFSA protein reacts directly with H2O2, and then consider the effects of this. In addition, in this experiment, it is also necessary to use standard substances known to have anti-fenton reaction effects, such as ascorbic acid, to evaluate the antioxidant activity in a relative manner. 

Answer 2: As you rightly observed, if the protein sample is directly exposed to a high concentration of H2O2 solution, the two will undoubtedly react. In this experiment to generate •OH, we first added an equal volume and molar concentration of FeSO4 and H2O2. According to the reaction equation, they interacted with each other in a ratio of 1:1 to generate •OH, so there was no residual H2O2 left upon the addition of the protein samples. 

Comments 3: The authors have evaluated the superoxide radical scavenging capacity using the xanthine oxidase-mediated superoxide radical production system and NBT, but it has been reported that NBT interacts with xanthine oxidase. Therefore, the authors need to confirm that the rFSA protein does not directly interact with NBT, and also need to evaluate the superoxide radical scavenging capacity using other indicators such as XTT. In addition, in this experiment, it is also necessary to evaluate the antioxidant activity relative to standard substances known to have anti-fenton reaction effects, such as ascorbic acid, using antioxidants such as Trolox and SOD. 

Answer 3: We set up a blank control group in our experiment, i.e., protein sample solvent (PBS) and each reaction reagent solution (xanthine, xanthine oxidase, NBT), which should deduct the reaction of NBT and xanthine oxidase. After incubating the protein sample with NBT, the color of the NBT solution (related to its internal functional structure tetrazole) did not change significantly, and it was showed that they would not react chemically. Meanwhile, both the NBT and reagent XTT are tetrazole derivatives and they have a comparable reaction mechanism and reaction products (formazan) with superoxide anion.

Comments 4: There are many typos and some abbreviations, the full name of which is not mentioned in the manuscript at the first appearance.  The authors should carefully check the typos and the grammar thoroughly.

Answer 4:  Thank you for your suggestion. The mentioned concerns were revised in the manuscript with the revision marks.

Reviewer 2 Report

Comments and Suggestions for Authors

In this work, the authors,  with the main objective to application in pet foods and veterinary clinics, expressed and characterized recombinant feline serum albumin using Pichia pastoris reaching 3.89 mg/mLfrom 5-L fermenter level and the in vivo safety assay of  in mice revealed that this feline serum albumin seems to be safe. The manuscript is clear and detailed, the procedures and the results are well described, and the conclusions follow the achieved results, therefore, it can be considered for publication if the following minor revision is considered:

-        Please change the scales of Figure 5(A). Use linear scale or log scale for the concentration and for the hemodialysis place a scale between -2 and 2(%) for example. Please explain better the reason of the hemodialysis percentage to be negative. 

Author Response

Comments: Please change the scales of Figure 5(A). Use linear scale or log scale for the concentration and for the hemodialysis place a scale between -2 and 2(%) for example. Please explain better the reason of the hemodialysis percentage to be negative. 

Answer: Thank you for your suggestions. The the scales of Figure 5A has been changed in the manuscript. For the horizontal coordinates of this graph, we have used the log2 scale to even out the spacing. The negative value of the hemolysis rate of the sample might be attributed to the slightly hemolytic effect resulting from the improper control of the operation intensity when dealing with the blank control group. Additionally, the hemolysis test was repeated, and the revised Figure 5A has been uploaded.